# Determinants of Accepting or Rejecting Influenza Vaccination—Results of a Survey Among Ligurian Pharmacy Visitors During the 2023/2024 Vaccination Campaign

**DOI:** 10.3390/vaccines13060580

**Published:** 2025-05-29

**Authors:** Daniela Amicizia, Silvia Allegretti, Federico Grammatico, Matteo Astengo, Francesca Marchini, Alberto Battaglini, Irene Schenone, Irene Schiavetti, Camilla Sticchi, Barbara Rebesco, Filippo Ansaldi

**Affiliations:** 1Ligurian Regional Health Service, Piazza della Vittoria 15, 16121 Genoa, Italy; daniela.amicizia@unige.it (D.A.); federico.grammatico@alisa.liguria.it (F.G.); matteo.astengo@alisa.liguria.it (M.A.); francesca.marchini@alisa.liguria.it (F.M.); alberto.battaglini@alisa.liguria.it (A.B.); irene.schenone@alisa.liguria.it (I.S.); camilla.sticchi@asl4.liguria.it (C.S.); filippo.ansaldi@alisa.liguria.it (F.A.); 2Department of Health Sciences, University of Genoa, 16132 Genoa, Italy; irene.schiavetti@unige.it

**Keywords:** influenza, vaccination, pharmacies, hesitancy, vaccine uptake

## Abstract

**Background/Objectives:** Seasonal influenza vaccination is crucial for reducing morbidity, mortality, and healthcare burdens. The 2023/2024 Ligurian vaccination campaign (Italy) utilized an inclusive model involving local health authorities, general practitioners, pediatricians, and pharmacies to enhance accessibility. Our study aimed at focusing on factors influencing vaccine uptake, public attitudes and access to preventive healthcare services. **Methods:** A cross-sectional survey was conducted among adults (≥18 years) in Ligurian pharmacies visitors during the vaccination campaign. A self-administered structured questionnaire gathered data on demographics, vaccination history, healthcare access, and awareness. **Results:** The study included 30,499 participants, and the median age with P25–P75 (years) was 62.0 [47.0–74.0]; 54.6% were female. Considering determinants of accepting influenza vaccination, age was identified as a strong independent predictor. Each one-year increase in age was associated with a 3.8% increase in the odds of influenza vaccination (OR 1.03, 95% CI 1.03–1.04, *p* < 0.001). Compared to individuals who never visited their general practitioners, those who visited “sometimes”, “often”, or “very often” had significantly higher odds of influenza vaccination (OR 1.54, 1.97, and 1.98, respectively; *p* < 0.001 for all categories). The strongest predictor of influenza vaccination in the 2023/2024 season was having received the influenza vaccine in the previous season (2022/2023) (OR 71.73, 95% CI 65.38–78.78, *p* < 0.001). Consistent with increasing age predicting higher influenza vaccination uptake, older age was associated with lower odds of refusing the vaccine due to the belief that “getting or transmitting influenza does not matter” or due to “other or unspecified reasons”. In contrast, receipt of the COVID-19 vaccination significantly increased the odds of holding these opinions. Among individuals who cited reasons such as fear of side effects, concerns about vaccine safety, fear of injections, general opposition to vaccines, or doubts about vaccine effectiveness, having received the COVID-19 vaccine was associated with lower odds of citing these as barriers to influenza vaccination. **Conclusions:** Fear of side effects and perceived unnecessary vaccination are key barriers. Targeted education and the involvement of general practitioners could enhance vaccine acceptance, particularly among hesitant groups.

## 1. Introduction

Influenza is a respiratory disease resulting from infection with the influenza virus [1]; it represents a serious public health issue, causing a high epidemiological and clinical impact among populations and a significant source of cases to be managed, and of indirect costs due to workdays lost, care obligations for family members and unenjoyed recreational/sports activities [2,3,4,5,6].

Individuals can experience influenza multiple times during their lifetime, regardless of lifestyle, age and place of residence. Older adults, young children, pregnant women and persons with chronic illnesses are more susceptible to severe forms requiring hospitalizations. Serious complications include pneumonia, myocarditis, and encephalitis, which can lead to death. Influenza-related morbidity and mortality are greatest among the elderly [6]. The overall estimated mortality rate related to influenza is 13.8 deaths per 100,000 people each year in Europe. During the 2023/2024 winter season an increase in mortality was observed among adults aged ≥45, coinciding with the widespread circulation of respiratory viruses, in EU countries [7].

In Italy, the results of a recent systematic review suggest that influenza contributes to a significant excess in mortality, with rates among the elderly estimated to be over six times higher than in the general population. Indeed, the majority of influenza-related deaths (65–96%) occur in individuals aged 65 and older [8].

During the winter season 2023/2024, the incidence of Influenza Like Illness (ILI) in Italy reached its epidemic peak during week 52 of 2023, with an incidence rate >18 cases per 1000 inhabitants [9].

Seasonal influenza vaccination is a cornerstone of public health strategy to reduce morbidity and mortality, especially among high-risk groups [10,11]. Vaccines not only provide individual protection but also reduce transmission within communities, minimizing complications and healthcare system impact.

Despite the real value and benefits of influenza vaccination at the healthcare, social and economic level, unsatisfactory coverage is registered in many European countries, Italy included, with relevant repercussions for healthcare and society.

The last survey carried out by ECDC provided an update on vaccination coverage rates in EU/EEA Member States for the 2015–2016, 2016–2017 and 2017–2018 influenza seasons, reporting that none of the EU Member States achieved the recommended vaccination coverage target [12].

In Italy, annual seasonal influenza vaccination campaigns are structured initiatives designed to offer free vaccinations to the target populations recommended by the Ministry of Health’s annual circular. These campaigns also implement systems to effectively monitor and evaluate vaccination coverage [9]. To address the challenges posed by influenza, its associated complications, and its economic burden, a vaccination rate exceeding 75% is recommended for populations at high risk of severe diseases and for categories with older age [9,13].

Influenza vaccination coverage among the elderly in Italy has shown significant variations over the years, with fluctuating trends reflecting both the effectiveness of vaccination campaigns and public perception of the importance of vaccination.

A decline was recorded in the 2023/2024 season, with coverage dropping to 53.3%, marking a decrease of 3.4 percentage points compared to the previous season [14].

Therefore, it is essential to intensify awareness campaigns and improve access to vaccination in order to increase vaccination coverage and ensure greater protection against seasonal influenza.

Understanding knowledge, attitudes and practices dynamics regarding influenza vaccinations is crucial for refining regional strategies and institutional approaches, in order to enhance vaccination coverage rates. In this context, our study aimed at focusing on factors influencing vaccine uptake, public attitudes and access to preventive healthcare services.

## 2. Materials and Methods

### 2.1. Setting

The Liguria Region (Italy) has 1,507,636 inhabitants and healthcare is organized into five Local Health Units (LHUs) numbered from 1 to 5 from west to east; LHU3 encompasses the territory of the capital city of the region (Genoa) [15]. The catchment area is the following: LHU1 203,710; LHU2 258,713; LHU3 656,517; LHU4 135,852; and LHU5 209,044 subjects, respectively [16].

### 2.2. Seasonal Influenza Vaccine Campaign

The 2023/2024 regional seasonal influenza vaccination campaign was designed in line with Health Ministry recommendations, and planned by the Regional Authority, A.Li.Sa. in collaboration with the Regional Vaccination Commission and the LHUs. The free offer of influenza vaccination is guaranteed by the Prevention Departments of LHUs, general practitioners (GPs), pediatricians and community pharmacies [16]. This organizational structure was planned to enhance coverage through an inclusive and accessible model.

The vaccination campaign took place from October 2023 to January of the following year, with most activities concentrated between the months of October and December.

### 2.3. Survey

The survey was carried out during the 2023/2024 influenza vaccination campaign in Ligurian community pharmacies by means a self-administered structured questionnaire. Adults aged ≥18 who used pharmacy services were invited to participate voluntarily. Participants completed a structured questionnaire addressing demographics, vaccination history, healthcare access, and knowledge of other preventive services.

The data were digitally collected during the influenza vaccination campaign (October 2023–December 2023).

The structured questionnaire included several items with dichotomous (Yes/No) and open-ended formats. It captured the following:−Demographics: age, gender, education level and occupation (for working-age individuals).−Vaccination History: influenza vaccination status during the 2022/2023 season, intention to vaccinate in the 2023/2024 season, COVID-19 vaccination history, and reasons to receiving or not receiving the flu vaccine in the previous season (if applicable).−Awareness of Preventive Health Services: knowledge of oncologic screening (e.g., mammography, PAP/HPV testing, fecal immunochemical test (FIT), as well as adult vaccinations (e.g., pneumococcal, herpes zoster, tetanus boosters).−Healthcare Access and Utilization: participants’ experiences accessing healthcare services (e.g., GPs, specialists, nurses) over the past year and their preferred contact options for health needs, such as pharmacies, GPs, emergency rooms, public clinics and private facilities.

### 2.4. Statistical Analyses

The collected data are presented as means with standard deviations or counts with frequencies (%), as appropriate. A univariate logistic regression analysis, followed by a multivariate model, was conducted to assess the factors associated with influenza vaccine intent or uptake during the 2023/2024 season.

In order to identify factors associated with different reasons for not receiving the influenza vaccine, five binary logistic regression models were fitted, each corresponding to one of the five predefined outcomes (Reason 1 to Reason 5).

For each outcome, univariate logistic regression analyses were first performed separately for the following independent variables: sex, age, level of education, frequency of contact with the general practitioner, frequency of pharmacy visits, ease of healthcare access, receipt of COVID-19 vaccination, and receipt of influenza vaccination in the 2022/2023 season.

Subsequently, multivariate logistic regression models were constructed for each outcome, including all independent variables simultaneously to adjust for potential confounding.

Odds ratios and 95% confidence intervals were reported for all models. In cases where the logistic regression model encountered quasi-complete separation (e.g., one category perfectly predicting the outcome), the odds ratio was not estimable and is reported as such.

To account for multiple comparisons across predictors and outcomes, *p*-values were adjusted using the Bonferroni correction. Both unadjusted and Bonferroni-adjusted *p*-values are reported. Statistical significance was defined as Bonferroni-adjusted *p*-values < 0.05.

Statistical analyses were performed using R version 4.1.3 (released on 10 March 2022).

## 3. Results

A total of 30,499 individuals participated to the study [54.6% females (*n* = 16,662) and 45.4% males (*n* = 13,837)]. The percentage of survey refusals was 9%. The median age with P25–P75 (years) was 62.0 [47.0–74.0]. The distribution of age class expressed in quartile broken down by the 5 LHUs of Liguria (LHU 1, LHU 2, LHU 3, LHU 4, LHU 5) is reported in Table 1.

Regarding the education level of participants, the responses showed that 34.1% had completed secondary high school (4–5 years), followed by 26.3% with a university degree, 12.6% high school (2–3 years), 8.2% held elementary school and 18.8% middle school, respectively.

Employment status revealed that 57.6% were employed, with the remainder retired or unemployed. The most frequent type of profession was employee followed by freelancer.

Among the study participants, 60.8% reported having received the influenza vaccine during the 2022/2023 season and, regarding the 2023/2024 vaccination season, 62.6% expressed their intention to be vaccinated or confirmed that they had already been vaccinated (Table 2).

Overall, 71.4% reported having at least some contact with a GP, while 89.1% referred pharmacy attendance. Criticism regarding contact with healthcare professionals was expressed by 29.3% of respondents.

Regarding the COVID-19 vaccination, 89.2% of respondents (*n =* 27,209/30,499) reported accepting the vaccine offer.

Awareness of pneumococcal and herpes zoster vaccines and tetanus booster vaccines recommended for adults, elderly and individual at risk, was high, with 79,1% (*n* = 24,134) of participants informed about these prevention measures.

The survey also explored secondary prevention opportunities, such as oncologic screening. A total of 76.3% of respondents (*n*= 23,267) indicated they were aware of public health screening programs recommended by age and offered free of charge.

In relation to the frequency of contact with general practitioners, 34.4% of participants declared “sometimes”, 24.2% “often”, and 12.8% “very often” (Table 2).

Regarding pharmacies, 33.6% of the enrolled individuals affirmed “often”, 29.6% “very often”, and 25.9% “sometimes”, respectively. Concerning the question on the ease of contacting a healthcare professional during the past year, 9.3% of respondents reported it was “very easy”, while 24.0% found it “fairly easy”, and 37.4% “neutral”.

Three distinct items were investigated: adherence to influenza vaccination, COVID-19 vaccination and compliance to previous seasonal influenza vaccination. The results show that 62.6% were already vaccinated with the influenza vaccine for the current season or planned to receive it, 60.8% had received influenza vaccine in the 2022/2023 season and 89.2% had received at least one dose of COVID vaccine during their life (Table 3).

In relation to the responses of those who had been vaccinated or intended to be vaccinated (19,872 individuals), the primary motivation was personal protection against the risk of infection (*n* = 11,861, 63.3%). Additionally, 740 participants (4.0%) were motivated by endorsements related to work, gym attendance, or university activities (Figure 1).

The primary reasons for not being vaccinated (*n* = 10,627) varied (Figure 2), with the most common concern referring to a low perceived severity of influenza, reported by 17.3% of the unvaccinated group, followed by potential side effects, which accounted for 15.7% of the total population.

A lack of time to be vaccinated was another significant barrier, mentioned by 7.4% of unvaccinated respondents. Fear of injections also played a role, affecting 3.8% of those who did not become vaccinated. Additionally, concerns about vaccine safety influenced decisions, with 6.7% of unvaccinated individuals believing the vaccine to be dangerous, 7.8% who declared that the vaccine was not effective (Figure 2) and 10.9% declaring themselves to be anti-vaccination.

Appendix A shows the factors associated with influenza vaccine intent or uptake in the 2023/2024 season using univariate and multivariate analysis. We observed that the variables age, visit to GPs, compliance with previous influenza vaccine offer and COVID vaccine were positively and statistically significantly associated with the willingness to be vaccinated. Specifically, the multivariate analysis demonstrated a positive correlation between medical visits conducted by GPs and the likelihood of vaccination, showing a gradual increase as the frequency of visits increased. Additionally, individuals who had been vaccinated in the previous season showed a higher probability of being vaccinated in the study season, as well as previous immunization with COVID-19 vaccine. Age also emerged as a statistically significant factor.

In contrast, sex, education level, frequency of pharmacy visits and perceived ease of accessing healthcare services were not significantly associated with influenza vaccination in the multivariate model.

Appendix A and Table 4 present the factors associated with reasons for not receiving the influenza vaccine, using univariate and multivariate analysis, respectively.

It can be observed that, regarding reason 1, “Getting the flu or transmitting it does not matter”, the predictor “age” was a negative significative predictor [OR 0.99, *p* < 0.001], as opposed to “receipt of COVID-19 vaccination” (OR 4.41, *p* < 0.001).

For reason 2 (Fear of side effects/vaccine dangerous/afraid of injection), reason 3 (Against vaccine), and reason 4 (Vaccine not effective), a statistical significance (OR < 1, *p* < 0.001) was found for receipt of COVID-19 and influenza vaccines.

As regards reason 5 (All other and no reason), predictors such as age and often visit GP were found to be statistically significant (OR < 1, *p* < 0.001); healthcare contact (somewhat easy) and receipt of COVID-19 vaccine presented an OR > 1 (*p* < 0.001).

## 4. Discussion

Vaccination remains one of the most effective public health measures, recognized by the World Health Organization (WHO) as a key element of the human right to health and a shared responsibility among individuals, communities and governments. Influenza vaccination, in particular, plays a critical role in protecting children, adults and the elderly, with uptake influenced by a mix of personal, structural, and informational factors [1,17,18].

Our study explored the vaccination intentions of survey participants, offering valuable insights into public attitudes within a large population. The findings reveal a complex interplay of factors influencing vaccine uptake.

Notably, individuals who received the influenza vaccine in the 2022/2023 campaign were more likely to adhere in the subsequent season.

On the contrary, Schmid et al. (2017) [19] in their systemic review found, in nine studies targeted to the elderly, that previous influenza vaccination strongly influenced the decision to be vaccinated in subsequent seasons. Comparable findings were reported in the study by Klett-Tammen et al. [20].

Focusing on subject non-adherence to seasonal vaccination campaigns, number of visits to a GP was not a positive predictor. This could be explained by the fact that low vaccination rates among healthcare providers pose significant risks not only to themselves but also to their patients [21].

Nevertheless, these findings underscore the importance of continuity in preventive healthcare: individuals with positive vaccination experiences or regular interactions with healthcare providers are more likely to adhere to immunization programs [21,22].

A recent systematic review evaluating the effectiveness of general practitioners’ involvement in vaccination campaigns observed that actions carried out by GPs to reach the adult population were effective. GPs were revealed to be the key partners in public health efforts to improve vaccination coverage [23].

A strategy to implement satisfatory vaccination rates could be the implementation of clinical reminders and the promotion of direct conversations about vaccination within GPs’ offices, which have proven to be effective strategies to increase vaccination rates in adults, with interventions showing a 4% to 42% increase, depending on the intensity of the intervention [24].

Despite this encouraging evidence, vaccine hesitancy remains a critical barrier to public health efforts. Fear of side effects, the difficulty in finding time to become vaccinated, and perceived low risk of infection and disease were among the primary reasons for refusal, aligning with previous research that identified misinformation and safety concerns as key drivers of hesitancy [25,26]. A recent scoping review identified the most common determinants of vaccine hesitancy as a lack of confidence in vaccines and fear of potential adverse events [27].

We observed that the variable ‘age group’ was one of the predictive values, as reported by other authors. The intention to receive an influenza vaccination increased with increasing age [28]. Interestingly, when each of the five reasons for not being vaccinated was evaluated individually, a correlation emerged that suggested the opposite conclusion (OR < 1 for reasons 1 and 5).

Notably, we did not find an association between intention to receive influenza vaccination and educational level. This result could be attributed to the fact that older individuals, who have been the primary targets of seasonal vaccination campaigns for many year, tend to have lower educational attainment but higher vaccination rates compared to younger, more educated populations [29].

Capodici et al. found that inadequate knowledge regarding belonging to a target group was significantly associated with vaccine refusal or delayed acceptance. Other contributing factors to refusal included being female, aged 45–54, living in rural areas, having no higher education, perceiving vaccines as unsafe, and knowing individuals opposed to vaccination. Correcting these misconceptions and improving awareness may help increase vaccination uptake and reduce the overall impact of disease [30].

In our study the role of pharmacies as accessible vaccination points did not emerge as a significant factor in improving vaccine coverage. On the other hand, previous studies support the effectiveness of pharmacy-based interventions, such as proactive vaccination status checks and personalized recommendations [31,32]. For example, a systematic review by Murray et al. (2021) found that these interventions increased influenza vaccination rates by 24% compared to standard care [33].

However, our findings reinforce the potential of pharmacies as key vaccination hubs, particularly within the “Pharmacy of Services” model, pharmacied providing 20% of vaccine administration providers in the Liguria region. The other providers are GPs, paediatriacians, and prevention departments of LHUs. This approach foresees expansion of the role of pharmacies beyond medication dispensing, positioning them as community health centers that offer vaccinations and preventive care. Pharmacists have the potential to enhance patient care by improving treatment, adherence and outcomes related to influenza vaccinations. The active engagement of pharmacists could be crucial for the effective execution and success of prevention initiatives.

Nowadays, in Italy, there is an ongoing debate about expanding the role of pharmacists and increasing their involvement in administering vaccines—not just the flu shot—which could lead to higher vaccination rates.

Finally, the importance of continuous monitoring of vaccine hesitancy and strategic efforts is highly relevant and this issue is emphasized in the Italian National Vaccination Prevention Plan 2023–2025 (PNPV) [34]. The plan highlights that “communication in the vaccination field must today take into account a multiplicity of objectives, recipients, channels and methods through which it must be implemented, with the main purpose of building and maintaining the population’s trust in vaccinations and health institutions”. In line with this, cognitive surveys play a fundamental role in guiding personalized communication strategies. To address gaps in vaccination coverage, particularly among younger populations and those with limited healthcare access, targeted campaigns are essential. Digital tools, such as mobile apps and automated reminders, could facilitate proactive engagement and simplify vaccination scheduling [35,36]. Indeed, it was proven that reminders and recall systems for patients in primary care settings are likely to be effective in increasing the proportion of the target population receiving immunizations [37].

Moreover, strengthening collaborations among healthcare providers, including general practitioners, pharmacists, and specialists, is essential to identifying at-risk individuals and delivering personalized recommendations.

Notably, the analysis of reasons for not being vaccinated reveals several interesting insights. While easy access to healthcare generally appears to reduce fear and hesitation toward vaccination, Reason 5 (all other and no reason given) is more often chosen by individuals for which contact with healthcare was easy. However, contact with a GP reduced the odds for Reason 5, indicating that other healthcare providers may increase general doubts concerning influenza vaccination.

Since the patterns observed for Reasons 1 (getting flu or transmitting it does not matter) and 5 are similar; these concerns are more specific to influenza vaccination, while Reasons 2 (fear of side effects) and 4 (vaccine not effective) appear to reflect more general vaccine hesitancy. Therefore, effective public health campaigns should address both aspects: vaccination benefits in general and specifically the flu vaccination.

Despite these insights in our research, some limitations must be acknowledged. The use of a self-administered structured questionnaire allowed investigation of several matters, from prevention to use of heathcare services and satisfaction of users. However, several items, e.g., nationality, chronic disease and pregnancy, were not investigated and represent additional constraints. Furthermore, the survey did not inquire about risk factors for experiencing severe influenza upon infection, although this would have been possible and would not have increased the duration of filling in the questionnaire by very much time.

Potential language barriers could be present; however, the pharmacists were able to support the users to better understand the questions, if necessary. Patients with limited mobility, who may have difficulty accessing pharmacies, may also be underrepresented among the enrolled subjects.

Futhermore, self-reported data may also introduce recall bias, and actual vaccination rates may differ from self-reported responses. A check on regional vaccination registries was not performed.

A key strength is the large survey sample, providing a foundation for developing communication initiatives and promoting targeted vaccination campaigns based on age groups. However, the sample was not representative of the entire regional population (S2).

## 5. Conclusions

The study analyzed the main determinants of vaccination, with the aim of improving existing vaccination strategies and implementing concrete actions to increase vaccine adherence. Understanding public attitudes and behaviors allows healthcare systems to design personalized and targeted interventions that can increase vaccination coverage.

Since prior influenza vaccination was a strong positive predictor for individuals who accepted the vaccine, but not for those who remained unvaccinated, efforts should focus on implementing tailored interventions that directly address distorted risk perceptions, fear of adverse effects and widespread distrust in vaccines. These targeted strategies should aim to educate and build confidence, particularly among those hesitant to be vaccinated. Healthcare professionals, particularly GPs, play a key role in this process due to their regular interactions with adults and the high value patients place on their recommendations [28].

Strengthening communication strategies and leveraging diverse healthcare access points will be essential in addressing vaccine hesitancy and improving immunization rates across all population groups. Indeed, there is a need for targeted communication strategies that address the specific concerns of more educated groups through evidence-based messaging.

## Figures and Tables

**Figure 1 vaccines-13-00580-f001:**
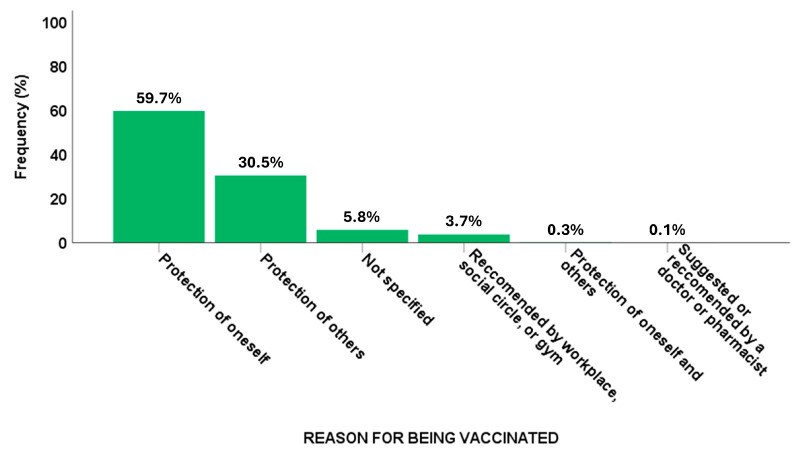
Reasons for flu vaccination.

**Figure 2 vaccines-13-00580-f002:**
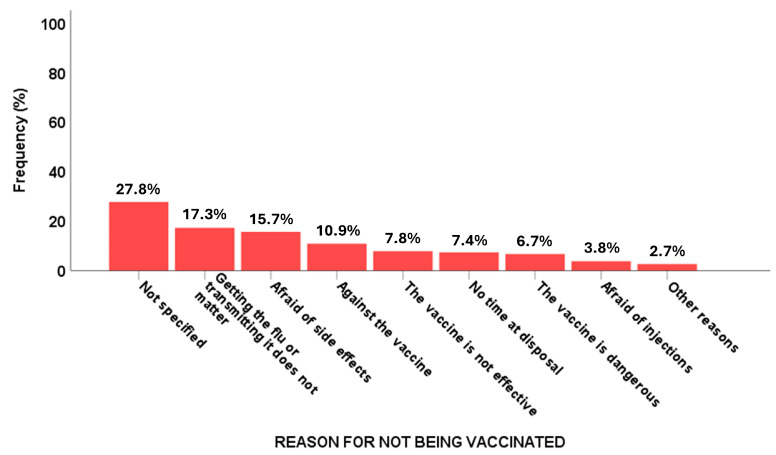
Reasons for not having flu vaccination.

**Table 1 vaccines-13-00580-t001:** Age distribution by healthcare districts (Local Health Unit, LHU).

Age Class	LHU 1	LHU 2	LHU 3	LHU 4	LHU 5	Total
18–47	1128 (23.8%)	1559 (24.0%)	3278 (27.2%)	689 (25.5%)	1116 (24.8%)	7770 (25.5%)
48–62	1172 (24.7%)	1638 (25.2%)	2910 (24.1%)	644 (23.8%)	1139 (25.4%)	7503 (24.6%)
63–74	1236 (26.1%)	1681 (25.9%)	2907 (24.1%)	723 (26.7%)	1077 (24.0%)	7624 (25.0%)
≥75	1200 (25.3%)	1613 (24.8%)	2977 (24.7%)	651 (24.0%)	1161 (25.8%)	7602 (24.9%)

**Table 2 vaccines-13-00580-t002:** Items related to medical services provided by healthcare professionals, investigated by means of questionnaires.

Item	Category	*n* (%)
Flu Vaccine 2022/2023 season uptake	Yes	18,533 (60.8%)
Flu Vaccine intent or uptake in the 2023/2024 season	Yes	19,094 (62.6%)
COVID-19 Vaccine uptake	Yes	27,209 (89.2%)
Aware that there are other vaccines recommended for adults (e.g., pneumococcal, herpes zoster, tetanus booster)	Yes	24,134 (79.1%)
Aware that age-specific screening tests are recommended and offered free of charge (e.g., mammogram, HPV test, FIT)	Yes	23,267 (76.3%)
Over the past 12 months, how often do you see your general practitioner (GP)?	Never	1693 (6.2%)
	Rarely	6080 (22.4%)
	Sometimes	9346 (34.4%)
	Often	6580 (24.2%)
	Very often	3479 (12.8%)
Over the past 12 months, how often did you go to the pharmacy?	Never	763 (2.8%)
	Rarely	2171 (8.1%)
	Sometimes	6991 (25.9%)
	Often	9054 (33.6%)
	Very often	7987 (29.6%)
Over the past 12 months, how easy has it been for you or a family member to contact a healthcare professional (e.g., GPs, specialists, nurses, etc.)?	Very difficult	2426 (8.0%)
	Somewhat difficult	6508 (21.3%)
	Neither easy nor difficult	11,416 (37.4%)
	Somewhat easy	7308 (24.0%)
	Very easy	2841 (9.3%)

**Table 3 vaccines-13-00580-t003:** Response to items related to vaccination: influenza vaccination in current and previous season, and COVID-19 vaccination.

Item	Reply	*n* (%)
Q1 *: Flu Vaccine intent or uptake in the 2023/2024 season	No	11,405 (37.4%)
	Yes	19,094 (62.6%)
Q2 *: COVID-19 Vaccine	No	3290 (10.8%)
	Yes	27,209 (89.2%)
Q3 *: Flu Vaccine uptake in the 2022/2023 season	No	11,966 (39.2%)
	Yes	18,533 (60.8%)
At least two answers of “yes” to the three previous questions (Q1, Q2, Q3)	No	10,627 (34.8%)
	Yes	19,872 (65.2%)

* Q = question.

**Table 4 vaccines-13-00580-t004:** Factors associated with reasons for not receiving the influenza vaccine. Results of multiple logistic regression with odds ratios (OR), 95% confidence intervals (CI), *p*-values and Bonferroni-adjusted *p*-value reported.

Multivariate Regression																
		Reason 1			Reason 2			Reason 3			Reason 4			Reason 5		
Predictor		OR 95%CI	*p* raw	*p* Bonf	OR 95%CI	*p* raw	*p* Bonf	OR 95%CI	*p* raw	*p* Bonf	OR 95%CI	*p* raw	*p* Bonf	OR 95%CI	*p* raw	*p* Bonf
Sex	Male vs. Female	1.17 (1.05–1.31)	0.004	0.391	0.88 (0.80–0.97)	0.008	0.815	1.10 (0.95–1.27)	0.194	0.99	1.06 (0.91–1.23)	0.456	0.99	0.96 (0.87–1.05)	0.327	0.99
Age (decade)		0.89 (0.86–0.93)	<0.001	<0.001	0.99 (0.96–1.03)	0.667	0.99	1.06 (1.01–1.11)	0.013	0.99	0.97 (0.93–1.02)	0.316	0.99	0.87 (0.85–0.90)	<0.001	<0.001
Education	Elementary school	Ref.			Ref.			Ref.			Ref.			Ref.		
	Middle school	1.43 (0.94–2.27)	0.109	0.99	0.90 (0.69–1.20)	0.480	0.99	1.42 (0.97–2.12)	0.077	0.99	0.77 (0.50–1.20)	0.228	0.99	1.25 (0.91–1.75)	0.182	0.99
	High school (2–3 years)	1.16 (0.75–1.85)	0.519	0.99	0.96 (0.72–1.28)	0.759	0.99	1.60 (1.08–2.40)	0.022	0.99	0.69 (0.44–1.10)	0.106	0.99	1.15 (0.83–1.63)	0.403	0.99
	High school (4–5 years)	1.43 (0.95–2.24)	0.097	0.99	0.87 (0.67–1.14)	0.306	0.99	1.24 (0.86–1.82)	0.257	0.99	0.86 (0.58–1.31)	0.465	0.99	1.15 (0.84–1.59)	0.389	0.99
	University degree	1.34 (0.89–2.10)	0.175	0.99	0.74 (0.57–0.98)	0.031	0.99	0.98 (0.67–1.46)	0.921	0.99	0.85 (0.57–1.30)	0.428	0.99	1.39 (1.02–1.93)	0.042	0.99
GPs	Never	Ref.			Ref.			Ref.			Ref.			Ref.		
	Rarely	0.98 (0.81–1.20)	0.867	0.99	1.03 (0.86–1.23)	0.746	0.99	1.21 (0.93–1.57)	0.154	0.99	1.01 (0.77–1.36)	0.920	0.99	0.85 (0.72–0.99)	0.041	0.99
	Sometimes	0.88 (0.72–1.08)	0.222	0.99	1.05 (0.88–1.26)	0.605	0.99	1.00 (0.77–1.31)	0.982	0.99	1.34 (1.01–1.79)	0.048	0.99	0.76 (0.65–0.90)	0.001	0.147
	Often	0.73 (0.58–0.93)	0.009	0.946	1.08 (0.88–1.34)	0.446	0.99	0.90 (0.66–1.23)	0.503	0.99	1.15 (0.82–1.61)	0.425	0.99	0.64 (0.52–0.77)	<0.001	<0.001
	Very often	0.84 (0.62–1.13)	0.250	0.99	1.06 (0.82–1.36)	0.654	0.99	0.91 (0.62–1.32)	0.616	0.99	1.23 (0.82–1.85)	0.315	0.99	0.62 (0.49–0.79)	<0.001	0.009
Pharmacy	Never	Ref.			Ref.			Ref.			Ref.			Ref.		
	Rarely	1.03 (0.77–1.40)	0.843	0.99	1.09 (0.84–1.42)	0.507	0.99	0.82 (0.57–1.19)	0.293	0.99	0.72 (0.49–1.06)	0.091	0.99	1.23 (0.96–1.59)	0.108	0.99
	Sometimes	1.11 (0.84–1.48)	0.461	0.99	0.89 (0.70–1.14)	0.364	0.99	1.19 (0.85–1.68)	0.313	0.99	0.79 (0.57–1.13)	0.184	0.99	1.12 (0.88–1.42)	0.359	0.99
	Often	1.06 (0.80–1.41)	0.708	0.99	0.93 (0.73–1.19)	0.545	0.99	0.99 (0.70–1.42)	0.963	0.99	0.73 (0.52–1.05)	0.085	0.99	1.27 (1.00–1.62)	0.049	0.99
	Very often	1.08 (0.80–1.46)	0.618	0.99	1.04 (0.80–1.34)	0.789	0.99	0.85 (0.59–1.23)	0.380	0.99	0.51 (0.35–0.76)	0.001	0.063	1.55 (1.21–1.99)	0.001	0.054
Healthcare contact	Very difficult	Ref.			Ref.			Ref.			Ref.			Ref.		
	Somewhat difficult	1.12 (0.88–1.43)	0.348	0.99	0.92 (0.76–1.11)	0.360	0.99	0.84 (0.64–1.10)	0.204	0.99	0.99 (0.74–1.33)	0.924	0.99	1.17 (0.96–1.43)	0.124	0.99
	Neither easy nor difficult	1.12 (0.90–1.41)	0.309	0.99	0.87 (0.73–1.03)	0.111	0.99	0.86 (0.67–1.11)	0.235	0.99	0.85 (0.65–1.13)	0.250	0.99	1.39 (1.16–1.68)	<0.001	0.037
	Somewhat easy	1.25 (0.99–1.58)	0.063	0.99	0.80 (0.67–0.97)	0.021	0.99	0.79 (0.61–1.05)	0.098	0.99	1.01 (0.76–1.36)	0.936	0.99	1.57 (1.30–1.90)	<0.001	<0.001
	Very easy	1.26 (0.97–1.65)	0.090	0.99	0.65 (0.52–0.82)	<0.001	0.028	1.11 (0.81–1.53)	0.520	0.99	0.83 (0.58–1.19)	0.305	0.99	1.50 (1.21–1.87)	<0.001	0.026
COVID-19 vax	Yes vs. No	4.41 (3.59–5.49)	<0.001	<0.001	0.24 (0.22–0.27)	<0.001	<0.001	0.10 (0.08–0.11)	<0.001	<0.001	0.58 (0.49–0.69)	<0.001	<0.001	6.22 (5.26–7.39)	<0.001	<0.001

Reason 1 = Getting the flu or transmitting it does not matter; Reason 2 = Fear of side effects/vaccine dangerous/afraid of injection; Reason 3 = Against vaccine; Reason 4 = Vaccine not effective; Reason 5 = All other and no reason given. Scholarity = level of education; GPs = frequency of contact with the general practitioner; Pharmacy = frequency of pharmacy visits, Healthcare contact = ease of healthcare access; COVID-19 vax = receipt of COVID-19 vaccination; Flu vaccination 2022/2023: Defined as receipt of the influenza vaccine during the 2022/2023 season. This factor was not included in the analysis as it was not estimable.

## Data Availability

The data are not publicly available due to privacy and ethical restrictions.

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
