# Peer review of "Determinants of Accepting or Rejecting Influenza Vaccination—Results of a Survey Among Ligurian Pharmacy Visitors During the 2023/2024 Vaccination Campaign"

_vaccines, 2025, doi:10.3390/vaccines13060580_

Round 1

Reviewer 1 Report (Previous Reviewer 1)

Comments and Suggestions for Authors

Dear Authors,

Thank you for resubmiting your article. Its results are important.

I still have some comments.

Abstract: When you name the region Liguria, please add the country in parentheses. Background/Objectives should ends with the aim of the study, so please add it. The sentence describing statistical software and the analyses should be omitted.

  1. Materials and Methods:

Statistical analyses: The previous version of your manuscript included the median therefore the distribution of the variables is not Gaussian. Please add an explanation why did you switch to the mean in this version. The age distribution of pharmacy visitors is usually skewed; therefore the median is better to be reported (together with the range or IQR). 

Author Response

The country name was added in parentheses when Ligurian Region was mentioned in abstract, as required.

  1. As suggested, the median was reported (together with the range). We agree  it is a better measure of the central tendency of the group as it is not skewed by exceptionally high or low characteristic values.

Reviewer 2 Report (Previous Reviewer 3)

Comments and Suggestions for Authors

I appreciate the substantial revision improvements made along the draft publication.

You have taken into serious consideration all of my comments and likely could not do better. The main methodology limitations have been clearly mentionned. 

The conclusions are relevant and fit the study results.

Thus, i accept your draft for publication

NB some editing in the tables may be required but maybe it is due to errors of connections on my laptop? please check again

Author Response

We thank the reviewer for the comments. As required, editing of tables have been carried out.

Reviewer 3 Report (Previous Reviewer 4)

Comments and Suggestions for Authors

The MS “Determinants of accepting or rejecting influenza vaccination – Results of a survey among Ligurian pharmacy visitors during the 2023/24 vaccination campaign” is a much improved version of the originally rejected MS submitted by Amicizia et al. However, there are still a number of minor and major shortcomings. The greatest deficiency is the absence of a meaningful analysis of the determinants of rejecting the flu vaccination. As the primary goal of the study was to give recommendations how to improve vaccination coverage this is particularly disconcerting given the fact that such an analysis could easily been done as explained below.

General comments:

You use various versions to refer to flu seasons. Please stick to one, preferably 2022/23 etc.

Legends of figures should be below and not above the figure. Please show the percentages in the columns (or above the columns) in these charts.

All table legends are insufficient. Avoid abbreviations (Table 1) and precisely describe what the tables show. Results for the three questions about vaccination is surely no proper legend for a table.

Specific comments:

  1. Line 57: It are ‘inhabitants’ not ‘patients’

  1. Lines 90/91: I do not consider the name Local Health Unit a very good choice although it might be established in Italy. I would rather prefer to call the Healthcare Regions (HCR).

  1. Line 108: Describing the questionnaire is non-standardized is inappropriate and superfluous since there is no standardized questionnaire covering the topics of the inquiry (likewise, mentioning this as a limitation of the study is unnecessary, but you can mention as a limitation that you have not inquired about risk factors for experiencing severe influenza upon infection although this would have been possible and would not have increased the duration for filling in the questionnaire very much and if necessary you could have deleted knowledge of other precautionary opportunities offered free of charge that seems not very relevant). You can describe the questionnaire as self-administered structured questionnaire.

  1. Table 1: Change legend to: “Age distribution by healthcare districts (Local Health Unit, LHU)”

  1. Table 2: Change legend and add a header line with three columns “Item”, “Category”, “n (%)”

  1. Table 2: It is a bit surprising that 763 individuals stated they never visit a pharmacy although they were visiting a pharmacy when inquired.

  1. Table 2: Correct to ‘healthcare professional’

  1. Line 170-175: You should not and need not repeat what is already shown in a table. Point to the most important result or combine the results meaningfully. E.g. Overall, 71,4 % reported to have at least sometimes contact with a general practitioner.

  1. Line 206: Replace “willness” by “willingness”

  1. Table 4: Replace title by: “Factors associated with flu vaccine intent or uptake in the 2023/24 season. Results of univariate and multiple logistic regression with odds ratios (OR), 95% confidence intervals (CI) and p-values reported.”

  1. As mentioned above the intention to not get vaccinated should be analyzed directly applying the same methods as used for analyzing vaccine uptake intention. But in this case the dependent variables are the different reasons for not getting vaccinated. This would best be done by constructing 5 dummy variables: 1. Flu does not matter, 2. Fear of side-effects/vaccine dangerous/afraid of injections, 3. Against vaccines, 4. Vaccine not effective, 5- all other and no reason given. These 5 variables should be analyzed using the same independent variables as used for the vaccination intention. By this you would gain insight into the features of individuals that have these views, which would allow tailoring of information campaigns etc. Instead of using separate univariate and multiple logistic regression, all 5 dependent variables can be analyzed together in a multivariate logistic regression analysis, but this may be to complicated to interpret.

Author Response

Legends of figures have been inserted below the figure. As required, the percentages in the columns have been added in these charts.

As suggested, all table legends have been revised.

Specific comments:

  1. Line 57: done
  2. Lines 90/91: We prefer using the term “local health unit (LHU)” that in the Italian context is a public organization responsible for delivering healthcare services within a specific geographic area. These units manage healthcare facilities, preventive programs, and hospital services, and are a key part of the National Health Service in Italy.  The term suggested is not correct for Italian contest.
  3. Line 108: Done
  4. Table 1: Done
  5. Table 2: Done
  6. Table 2: We agree with your comment. The term “never” was considered by participants “the last period (12 months)”.
  7. Table 2: Done
  8. Line 170-175: We changed the text and revised according the suggestion.
  9. Line 206: Done
  10. Table 4: Done
  11. We thank the reviewer for the suggestion. However, we believe that conducting five separate regression analyses, each focused on a different reason for not getting vaccinated, would shift the focus away from our primary research objective, which is to identify the factors associated with the intention to get vaccinated. From a methodological point of view, analysing the determinants of each individual reason for not getting vaccinated would require multiple parallel models with the same set of covariates. While technically feasible, this approach would imply a shift in focus and may lead to redundancy and a potential inflation of type I error due to multiple testing. Moreover, such analyses would require an orientation toward understanding motivational profiles, rather than vaccination behaviour. For these reasons, we would prefer to focus on the main outcome, which is the vaccination intention, while describing the reasons for non-vaccination in a descriptive manner.

Round 2

Reviewer 3 Report (Previous Reviewer 4)

Comments and Suggestions for Authors

The revised MS “Determinants of accepting or rejecting influenza vaccination – Results of a survey among Ligurian pharmacy visitors during the 2023/24 vaccination campaign” has been much improved although there are still some minor errors. However, the biggest disadvantage is the authors’ reluctance to carry out an analysis of the reasons to not get vaccinated, as explained below their arguments are not convincing.

  1. Lines 137/138: What do the ° signs indicate? And what reason is there to restrict computation of the median and IQR to the range 25-75? Compute these values for the total range of values since no distortion will be introduced by outliers.

  1. Table 2: If the question about pharmacy visits was understood to cover the past 12 months then you should mention this in the question, e.g. How often did you go to a pharmacy (last 12 months)? Otherwise the category ‘never’ seems odd.

  1. Table 3: Change title to: “Response to items related to vaccination: influenza vaccination in current and previous season, and Covid-19 vaccination”

  1. Figure 1: Change legend: “Reasons for getting flu vaccination”

  1. Figure 2: Change legend: “Reasons for not getting flu vaccination”

  1. Table 4: In its present form the table is not comprehensible. Increase the width by using the whole page widths. Take care that the entries in a column are not inappropriately broken. You can omit the brackets around confidence intervals.

  1. As mentioned above your answer to my recommendation about an analysis of the 5 mayor reasons for not getting vaccinated is not convincing. 1. It does not shift the attention from the main scope of the investigation. Those that received already the flu vaccine or intended to do so are without major interest because they need not be convinced to get vaccinated. Your major intention is to provide evidence for recommendation how to increase vaccination rates and this will have to focus on those that do not want to get vaccinated. 2. There is no issue with alpha inflation due to multiple testing. First, because you could apply a simultaneous analysis by multivariate logistic regression and second, if you apply 5 separate multiple regression analyses you can easily adjust for multiple endpoints. Since this analysis is more important than the analysis of the intention to get vaccinated, you can shift Table 4 to the Supplement. But preferentially a new Table 5 should be included with the results of the analysis of reasons for not getting vaccinated. This table should have one column for the independent variables and five columns for the five dependent variables. In each of these columns the OR and 95% CIs should be reported, because this is sufficient for judging the importance of the respective factors (p values can be computed from the CIs and ORs). If there is sufficient room for reporting the percentages as for Table 4 then do so.

Author Response

  1. Lines 137/138: What do the ° signs indicate? And what reason is there to restrict computation of the median and IQR to the range 25-75? Compute these values for the total range of values since no distortion will be introduced by outliers.

We thank the reviewer for the comment. The ° symbol indicate the 25th and 75th percentiles (e.g., 25° for the 25th percentile), but we acknowledge that this notation may be unclear. In the revised version, we will replace it with the notation P25 and P75 to avoid confusion. As for the statistical approach, we confirm that the median was calculated on the full dataset, without excluding any values. The 25th and 75th percentiles were reported to describe the interquartile range, which, by definition, represents the range between P25 and P75. No restriction of the data to the interquartile range was applied prior to computing these summary statistics.

2.Table 2: If the question about pharmacy visits was understood to cover the past 12 months then you should mention this in the question, e.g. How often did you go to a pharmacy (last 12 months)? Otherwise the category ‘never’ seems odd.

As suggested “over 12 months” was added.

3.Table 3: Change title to: “Response to items related to vaccination: influenza vaccination in current and previous season, and Covid-19 vaccination”.

Done

4. Figure 1: Change legend: “Reasons for getting flu vaccination”

Done

5.  Figure 2: Change legend: “Reasons for not getting flu vaccination”

Done

6.  Table 4: In its present form the table is not comprehensible. Increase the width by using the whole page widths. Take care that the entries in a column are not inappropriately broken. You can omit the brackets around confidence intervals.

Done

7. As mentioned above your answer to my recommendation about an analysis of the 5 mayor reasons for not getting vaccinated is not convincing. 1. It does not shift the attention from the main scope of the investigation. Those that received already the flu vaccine or intended to do so are without major interest because they need not be convinced to get vaccinated. Your major intention is to provide evidence for recommendation how to increase vaccination rates and this will have to focus on those that do not want to get vaccinated. 2. There is no issue with alpha inflation due to multiple testing. First, because you could apply a simultaneous analysis by multivariate logistic regression and second, if you apply 5 separate multiple regression analyses you can easily adjust for multiple endpoints. Since this analysis is more important than the analysis of the intention to get vaccinated, you can shift Table 4 to the Supplement. But preferentially a new Table 5 should be included with the results of the analysis of reasons for not getting vaccinated. This table should have one column for the independent variables and five columns for the five dependent variables. In each of these columns the OR and 95% CIs should be reported, because this is sufficient for judging the importance of the respective factors (p values can be computed from the CIs and ORs). If there is sufficient room for reporting the percentages as for Table 4 then do so.

Table 5 reports the 5 multivariate analyses using updated statistical methods.

The univariate analysis is reported in the appendix.

The discussion has been revised.

Round 3

Reviewer 3 Report (Previous Reviewer 4)

Comments and Suggestions for Authors

The revised MS “Determinants of accepting or rejecting influenza vaccination – Results of a survey among Ligurian pharmacy visitors during the 2023/24 vaccination campaign” has again been much improved and I have only requests for minor corrections that do not necessitate further review but can be checked and approved to the discretion of the editor.

  1. Abstract: Please give the ORs and CIs with two decimal digits only.

  1. Abstract: The text ‘Focusing…vaccine’ should be replaced by something like: “Consistent with increasing age predicting receipt of the influenza vaccination, not consenting to get it because of the opinion “getting influenza or transmitting it does not matter” or because of “other or unspecified reasons” increasing age reduced the odds while “receipt of Covid 19 vaccination” significantly increased the odds for these opinions. In contrast, for the other reasons not to get the flu vaccination (fear of side effects/vaccine dangerous/afraid of injection, being against vaccine, vaccine not effective) having received the Covid 19 vaccination reduced the odds for these reasons” (Note that these reasons fear etc. may apply to any vaccination and seem to define a certain attitude towards vaccination in general, the 1st and 5th reason, on the other hand, are more specific to influenza vaccination. It seems to follow that a targeted campaign must consider these very different groups)

  1. Table 4: Omit the line with flu vaccination in the season 2022/23 because not only for Reason 1 but for all other reasons as well the OR is not estimable (which Is to be expected). Inform in a footnote that this was the case.

  1. Table 4: Instead “Scholarity” use “Education”

  1. Table 4: The entries for “Age” should be changed. Due to the small standard error for age in years, the ORs and CIs cannot be used in the reported form. Give the ORs and CIs by an increase of 10 years. Write “Age (decade)”. You need not repeat the analysis but simple take instead of exp(ß) the ß values, multiply them by 10, do this for the confidence bounds of ß as well and transform these values by exp to get the ORs and CIs.

  1. Discussion: There are some interesting aspects that can be learned from the analysis of reasons not to get the vaccination. While easy contact with healthcare seems to protect against fear etc. of vaccination, the Reasons 5 (which are diverse) are more often put forward in those for which contact with healthcare was easy. But contact with a GP reduced the odds for Reason 5, indicating that other healthcare providers may increase general doubts concerning flu vaccination. Since the pattern is similar for Reasons 1 and 5, it seems these reasons are more focusing specifically on flu vaccination, while Reasons 2-4 do more generally apply to all vaccination. To target both aspects a campaign must address both, vaccination benefits in general and specifically flu vaccination.

Author Response

We thank reviewer comments that allowed to improve the revised manuscript.

Abstract: Please give the ORs and CIs with two decimal digits only.
As suggested the ORS and CI are reported with two decimal only.

Abstract: The text ‘Focusing…vaccine’ should be replaced by something like: “Consistent with increasing age predicting receipt of the influenza vaccination, not consenting to get it because of the opinion “getting influenza or transmitting it does not matter” or because of “other or unspecified reasons” increasing age reduced the odds while “receipt of Covid 19 vaccination” significantly increased the odds for these opinions. In contrast, for the other reasons not to get the flu vaccination (fear of side effects/vaccine dangerous/afraid of injection, being against vaccine, vaccine not effective) having received the Covid 19 vaccination reduced the odds for these reasons” (Note that these reasons fear etc. may apply to any vaccination and seem to define a certain attitude towards vaccination in general, the 1st and 5th reason, on the other hand, are more specific to influenza vaccination. It seems to follow that a targeted campaign must consider these very different groups)
The abstract has been revied according to reviewer suggestion.

Table 4: Omit the line with flu vaccination in the season 2022/23 because not only for Reason 1 but for all other reasons as well the OR is not estimable (which Is to be expected). Inform in a footnote that this was the case.
Done

able 4: Instead “Scholarity” use “Education”
Done

Table 4: The entries for “Age” should be changed. Due to the small standard error for age in years, the ORs and CIs cannot be used in the reported form. Give the ORs and CIs by an increase of 10 years. Write “Age (decade)”. You need not repeat the analysis but simple take instead of exp(ß) the ß values, multiply them by 10, do this for the confidence bounds of ß as well and transform these values by exp to get the ORs and CIs.

Done

Comment 6. Discussion: There are some interesting aspects that can be learned
from the analysis of reasons not to get the vaccination. While easy
contact with healthcare seems to protect against fear etc. of
vaccination, the Reasons 5 (which are diverse) are more often put
forward in those for which contact with healthcare was easy. But contact
with a GP reduced the odds for Reason 5, indicating that other
healthcare providers may increase general doubts concerning flu
vaccination. Since the pattern is similar for Reasons 1 and 5, it seems
these reasons are more focusing specifically on flu vaccination, while
Reasons 2-4 do more generally apply to all vaccination. To target both
aspects a campaign must address both, vaccination benefits in general
and specifically flu vaccination.

Done

This manuscript is a resubmission of an earlier submission. The following is a list of the peer review reports and author responses from that submission.

Round 1

Reviewer 1 Report

Comments and Suggestions for Authors

Dear Authors,

Your article is very interesting! I have several comments:

Abstract: When you name the region Liguria, please add the country in parentheses. Background/Objectives should ends with the aim of the study, so please add it. The sentence describing statistical software and the analyses should be omitted. Please add a measure of dispersion next to median age (line 19). It could be either the range (min and max) or IQR (25th and 75th percentile).

  1. Introduction: The first sentence is too long. Please split it.
  2. Materials and Methods: page 2, line 66: please add the country in parentheses.

2.4. Statistical analyses: Please add a sentence about the numerical variables. For example: The numerical variables were expressed as median and IQR.

  1. Results: page 3, line 13: Please add a measure of dispersion next to median age.

Figure 1: It would be better to base the chart on the proportions instead of the numbers.

Table 2: Some hidden numbers appear here, please recheck.

Table 4: Please add the leading 0 of the p-values. The presentation of your results should be consistent. The same for O.R and Odds ratio and CI, and p-value – name the columns the same way. I would suggest to write it as OR, no dots in it.

I did not see any tables illustrating chi square tests?

Author Response

  1. Introduction: The first sentence is too long. Please split it.

As required we did the suggested change.

  1. Materials and Methods: page 2, line 66: please add the country in parentheses.

Done

2.4. Statistical analyses: Please add a sentence about the numerical variables. For example: The numerical variables were expressed as median and IQR.

The statistical analyses have been redone and a sentence has been added.

  1. Results: page 3, line 13: Please add a measure of dispersion next to median age.

Figure 1: It would be better to base the chart on the proportions instead of the numbers.

Table 2: Some hidden numbers appear here, please recheck.

Table 4: Please add the leading 0 of the p-values. The presentation of your results should be consistent. The same for O.R and Odds ratio and CI, and p-value – name the columns the same way. I would suggest to write it as OR, no dots in it.

I did not see any tables illustrating chi square tests?

As suggested, we have performed statistical analysis as indicated by reviewer 4 and the presentation of data changed.

Reviewer 2 Report

Comments and Suggestions for Authors

Although, the topic of the paper is interesting, I have methodological concerns:

The introduction is quite short. 

Why is the second class of variable "age" too wide?

The authors present the median age instead on the mean age.  Why? Are there any outliers? How is age measured? 

The lines 164-165 are not clear. The type of analysis and the way it is performed are not mentioned (Table 2). The same is also true for the lines 172-174 (Table 3). 

Gender (1-0) (and not Male) is not statistically significant and it should not be included in the model (Table 4). In addition, the correct terms are not “inferior” and “superior” but “lower” and “upper”.

The authors did not perform a goodness of fit test, or other diagnostic tests to evaluate the model.

Comments on the Quality of English Language

No comments. 

Author Response

The introduction is quite short. 

We thank the review comments; the introduction has been improved.

Why is the second class of variable "age" too wide?

The distribution of age class is now expressed in quartile

The authors present the median age instead on the mean age.  Why? Are there any outliers? How is age measured? 

We modified the text reporting mean age. No outliers are present.

The lines 164-165 are not clear. The type of analysis and the way it is performed are not mentioned (Table 2). The same is also true for the lines 172-174 (Table 3). 

We modified this part of the manuscript.

We revised the methods used for statistical analysis.

Gender (1-0) (and not Male) is not statistically significant and it should not be included in the model (Table 4). In addition, the correct terms are not “inferior” and “superior” but “lower” and “upper”.

We agree with review comments and have revised accordingly.

The authors did not perform a goodness of fit test, or other diagnostic tests to evaluate the model.

We agree with reviewer comment and we performed new statistical analysis.

Reviewer 3 Report

Comments and Suggestions for Authors

The potential role of pharmacies in vaccination campaigns is an interesting public health issue. 

This draft paper could be published but has to be improved: both the presentation and the result discussion as well. 

  • The title should be shorter and indicate the region and country
  • The introduction should include references on the morbidity and mortality due to influenza in Italy and Liguria region, the influenza coverage in Italy and Liguria region, and the data on the decline in influenza vaccination coverage in European countries.
  • The methodology should indicate the nature of the non-standardized questionnaire, the information collected on the existence of chronic illnesses, and on pregnancies, the assistance provided for the understanding of the questionnaire (language?), and on the nationality identification
  • The results shoud indicate: 1)the pourcentage of survey refusal or non participation, 2) table should give percentages 3) table 2, 3 and 4 should be given close to the related text4) any results from pregnant women , patients with chronic illnesses, elderly
  • Discussions on the following: 1) survey representativity in terms of population structure, vaccination coverage, health status and access to health care 2) survey period: how long has been the vaccination campaign ? Any time factor on the questionnaire response  with the risk of higher hesitancy at the onset of the campaign?3) the difference between the participants response and the actual vaccination rates and why?4)any difference on hesitancy for other disease vaccination 5)Why the regional registries have not been checked?
  • Conclusions : The study limitations and strenghts have to be integrated into the discussion section.The conclusion should be consistent with the methodology used with a special attention to the role of pharmacies and with the specific survey findings
  • Last , there are also few typing errors

Author Response

The potential role of pharmacies in vaccination campaigns is an interesting public health issue. 

This draft paper could be published but has to be improved: both the presentation and the result discussion as well. 

  • The title should be shorter and indicate the region and country

We changed the title according to reviewer 4.

  • The introduction should include references on the morbidity and mortality due to influenza in Italy and Liguria region, the influenza coverage in Italy and Liguria region, and the data on the decline in influenza vaccination coverage in European countries.

On the basis of reviewer suggestions, we included references on the morbidity and mortality due to influenza in Italy as well as vaccination coverage rate. To note that mortality data due to influenza are lacking in Liguria.

  • The methodology should indicate the nature of the non-standardized questionnaire, the information collected on the existence of chronic illnesses, and on pregnancies, the assistance provided for the understanding of the questionnaire (language?), and on the nationality identification

We add the information of the questionnaire in methods section.

  • The results should indicate: 1) the percentage of survey refusal or non participation, 2) table should give percentages 3) table 2, 3 and 4 should be given close to the related text4) any results from pregnant women , patients with chronic illnesses, elderly

The percentage of refusal was about 9% and this information has been added. The tables have been changed. No information on pregnant women and individuals with chronic diseases was available.

  • Discussions on the following: 1) survey representativity in terms of population structure, vaccination coverage, health status and access to health care 2) survey period: how long has been the vaccination campaign ? Any time factor on the questionnaire response with the risk of higher hesitancy at the onset of the campaign?3) the difference between the participants response and the actual vaccination rates and why?4) any difference on hesitancy for other disease vaccination 5) Why the regional registries have not been checked?

We followed the study protocol and only anonymous data were collected, therefore no further checks are feasible.

  • Conclusions: The study limitations and strengths have to be integrated into the discussion section. The conclusion should be consistent with the methodology used with a special attention to the role of pharmacies and with the specific survey findings

We revised conclusions according to reviewer suggestions.

  • Last, there are also few typing errors

We checked and revised all main text.

Reviewer 4 Report

Comments and Suggestions for Authors

The MS “A Survey-based investigation on Determinants of Influenza Vaccination Among Users of pharmacies: Results from the 2023/2024 Campaign” submitted by Amicizia et al. reports results of a survey among Ligurian visitors of pharmacies about influenza vaccination and its barriers and factors improving vaccination uptake. Although the study methodology is questionable and the statistical analysis faulty, I do not recommend rejection because still the study could be evaluated in a way that provides useful insights into the decision about getting vaccinated. Details about how the study can be meaningfully evaluated are provided below.

  1. The title is misleading and insufficient. A suggest to change it as follows: “Determinants of accepting or rejecting influenza vaccination – Results of a survey among Ligurian pharmacy visitors during the 2023/24 vaccination campaign”

  1. Line 17: Write “…among adult (≥18 years) Ligurian pharmacy visitors”

  1. Line 24: Write “transmission to”

  1. Lines 25ff: The results section of the Abstract need to be improved after the analysis has been corrected.

  1. Line 36: Replace “hight” by “high”

  1. Lines 37/38: Write “and is a significant…”. Change the wording since there are no indirect costs of managing the disease, these are the direct costs. Suggestion: “…a significant source of direct costs for the health care system from managing cases and of indirect costs due to workdays lost, care obligations for family members etc.”

  1. Line 39: Write: “Individuals can experience influenza multiple times…”

  1. Line 50: Write: “In Italy, seasonal influenza vaccination campaigns…”

  1. Line 58: Write: “at the healthcare, social…”

  1. Line 66: There is no reason to capitalize “healthcare”

  1. Line 68: Write: “capital city (Genova)”

  1. Lines 75/76: Again, there is no reason to capitalize “general practitioners”, “pediatricians” etc. Only proper names need to be capitalized.

  1. Section 2.3: The greatest methodological shortcoming is organizing the survey in pharmacies. Visitors of pharmacies do not represent the general adult population. Hence, any inference from this sample into the general Ligurian population is biased. Therefore, two precautions have to be applied: first, no obtained percentage can be directly interpreted as reflecting something about the population; second, none of these figures can be directly used to delineate recommendations for future campaigns. However, while the frequencies of each of the obtained manifestations of the variables are biased, relationships between variables are not or less affected by the survey method. Therefore, you should concentrate on this aspect. Furthermore, you need to add a section about the limitations of your study in the Discussion section.

  1. Section 2.4. This section must be rewritten after having applied the methods I suggest below. Write “Level of statistical significance was set to 5%”

  1. Results: General remark: Do not use phrases such as “Table X shows…” or “as Table Y shows”. Just write what you want to say about the results summarized in a table (or figure) and then reference it by adding the item in brackets (e.g. …the highest proportion of participants was found in the age group … (Table 1))

  1. Table 1: Change the title to “Age distribution of survey participants recruited among Ligurian pharmacy visitors by local health authority (LHU)”. Provide row percentages in brackets after the n for all age categories. After the line with the total provide a further line with the percentages of the adult Ligurian population in the respective age categories.

  1. Line 117/118: Since you don’t provide a category 60+ in your table, rephrase the sentence. The age category with the highest percentage is 30-59! Furthermore, you cannot speak of ‘participation rate’ because you don’t know how many visitors of pharmacies in the respective age groups have not participated.

  1. Line 121: Delete “secondary”

  1. Line 127: Replace “get” by “got”

  1. Line 142: Replace “effectiveness” by “effective”

  1. Figure 1: Change the title to “Reasons provided by participants for rejecting influenza vaccination in the season 2023/24. Percentages given relative to all participants rejecting.”. Provide the percentages at the end of the bars and add the legend for the x axis (number of participants).

  1. Line 160ff: What are ‘primary professionals’?

  1. Table 2 and lines 164-171: The figures in the table make no sense. The dependent variable is the intention to get vaccinated or having already received vaccination. The independent variable is age. An odds ratio is the ratio of the odds to find a person with the intention to get vaccinated or having received the vaccination in a certain subgroup relative to the odds in a reference group. But what is the reference in this case? All age groups are reported with an odds ratio, which cannot be because then there would be no reference! Most likely authors have erroneously taken for each age group the combination of all other groups as reference. This violates the independency assumption and the analysis has to be repeated by taking one age group as reference. This should be the youngest age group because it seems that there is a steady increase with increasing age. To test this trend hypothesis, a trend test should additionally be done. Furthermore, report for each age category the number and percentages of participants intending to get vaccinated or having been vaccinated by adding an additional column after the ‘N’ column. The p-values should be reported in the last column.

  1. Table 3: The same error as mentioned for table 2 is also present in Table 3. In this case, ‘University degree’ can serve as reference category. Change the title “Intention to get the flu vaccination or having already been vaccinated in season 2023/24 by educational category”. Write ‘Educational category’ as column header for the first column, the second column has the header ‘N (column %)”. Note that you have provided the OR and CI values with a decimal comma instead of a decimal point. As mentioned for table 2, provide also for table 3 the absolute numbers and percentages (for all educational categories) of those vaccinated but differentiate the percentages by writing for this column ‘vaccinated n (row %)’. The p-values should be reported in the last column.

  1. Table 4: Change the title to: “Results of multiple logistic regression of the intention to get the flu vaccination in the season 2023/24 or having already received it on several dichotomized predictive factors (yes/no refers to their presence or absence, respectively)”. Omit the useless footnote and the indication (1-0) and add four columns with the following entries: Yes n|Yes % vaccinated|No n|No % vaccinated. The yes/no refers to the variable in the line (e.g. Males). Report the CI in the form x-y and not in two columns. Include also age (below and above 60) and education (less or above high school level). The p-values should be reported in the last column.

  1. In order to make this paper a meaningful contribution the following additional analyses should be done and reported. 1) A multivariate (or if your software cannot calculate it you can also do separate analyses for each of the dependent variables) logistic regression of the 5 main reasons not to get vaccinated (you may combine fear of side effects and the opinion the vaccine is dangerous, because this is more or less the same) on the same predictors as reported in Table 4. 2) A cluster analyses of the reasons to reject flu vaccination. There might be two or three patterns of opinions that may turn out to need different pathways and arguments to improve acceptance of vaccination.

Comments on the Quality of English Language

There are some errors of grammar and style mentioned in my report to the authors. otherwise the English is fine.

Author Response

1. The title is misleading and insufficient. A suggest to change it as follows: “Determinants of accepting or rejecting influenza vaccination – Results of a survey among Ligurian pharmacy visitors during the 2023/24 vaccination campaign”

 Done

2. Line 17: Write “…among adult (≥18 years) Ligurian pharmacy visitors”

 Done

3. Line 24: Write “transmission to”

 Done

4. Lines 25ff: The results section of the Abstract need to be improved after the analysis has been corrected.

 Done

5. Line 36: Replace “hight” by “high”

 Done

6. Lines 37/38: Write “and is a significant…”. Change the wording since there are no indirect costs of managing the disease, these are the direct costs. Suggestion: “…a significant source of direct costs for the health care system from managing cases and of indirect costs due to workdays lost, care obligations for family members etc.”

 Done

7. Line 39: Write: “Individuals can experience influenza multiple times…”

 Done

8. Line 50: Write: “In Italy, seasonal influenza vaccination campaigns…”

 Done

9. Line 58: Write: “at the healthcare, social…”

 Done

10. Line 66: There is no reason to capitalize “healthcare”

 Done

11. Line 68: Write: “capital city (Genova)”

 Done

12. Lines 75/76: Again, there is no reason to capitalize “general practitioners”, “pediatricians” etc. Only proper names need to be capitalized.

 Done

13. Section 2.3: The greatest methodological shortcoming is organizing the survey in pharmacies. Visitors of pharmacies do not represent the general adult population. Hence, any inference from this sample into the general Ligurian population is biased. Therefore, two precautions have to be applied: first, no obtained percentage can be directly interpreted as reflecting something about the population; second, none of these figures can be directly used to delineate recommendations for future campaigns. However, while the frequencies of each of the obtained manifestations of the variables are biased, relationships between variables are not or less affected by the survey method. Therefore, you should concentrate on this aspect. Furthermore, you need to add a section about the limitations of your study in the Discussion section.

We followed the reviewer’ suggestions.

14. Section 2.4. This section must be rewritten after having applied the methods I suggest below. Write “Level of statistical significance was set to 5%”

 Level of statistical significance was set to <0.001.

15. Results: General remark: Do not use phrases such as “Table X shows…” or “as Table Y shows”. Just write what you want to say about the results summarized in a table (or figure) and then reference it by adding the item in brackets (e.g. …the highest proportion of participants was found in the age group … (Table 1))

 Done

16. Table 1: Change the title to “Age distribution of survey participants recruited among Ligurian pharmacy visitors by local health authority (LHU)”. Provide row percentages in brackets after the n for all age categories. After the line with the total provide a further line with the percentages of the adult Ligurian population in the respective age categories.

 Done

17. Line 117/118: Since you don’t provide a category 60+ in your table, rephrase the sentence. The age category with the highest percentage is 30-59! Furthermore, you cannot speak of ‘participation rate’ because you don’t know how many visitors of pharmacies in the respective age groups have not participated.

 We thank the reviewer for the comment, this part has been changed..

18. Line 121: Delete “secondary”

 Done

19. Line 127: Replace “get” by “got”

 Done

20. Line 142: Replace “effectiveness” by “effective”

 Done

21. Figure 1: Change the title to “Reasons provided by participants for rejecting influenza vaccination in the season 2023/24. Percentages given relative to all participants rejecting.”. Provide the percentages at the end of the bars and add the legend for the x axis (number of participants).

As required, we changed figure 1 and 2. 

22. Line 160ff: What are ‘primary professionals’?

We delete primary professional, they are are professional belonging to out-of-hours medical service

23. Table 2 and lines 164-171: The figures in the table make no sense. The dependent variable is the intention to get vaccinated or having already received vaccination. The independent variable is age. An odds ratio is the ratio of the odds to find a person with the intention to get vaccinated or having received the vaccination in a certain subgroup relative to the odds in a reference group. But what is the reference in this case? All age groups are reported with an odds ratio, which cannot be because then there would be no reference! Most likely authors have erroneously taken for each age group the combination of all other groups as reference. This violates the independency assumption and the analysis has to be repeated by taking one age group as reference. This should be the youngest age group because it seems that there is a steady increase with increasing age. To test this trend hypothesis, a trend test should additionally be done. Furthermore, report for each age category the number and percentages of participants intending to get vaccinated or having been vaccinated by adding an additional column after the ‘N’ column. The p-values should be reported in the last column.

 We thank the observations; we applied the suggested methods.

24. Table 3: The same error as mentioned for table 2 is also present in Table 3. In this case, ‘University degree’ can serve as reference category. Change the title “Intention to get the flu vaccination or having already been vaccinated in season 2023/24 by educational category”. Write ‘Educational category’ as column header for the first column, the second column has the header ‘N (column %)”. Note that you have provided the OR and CI values with a decimal comma instead of a decimal point. As mentioned for table 2, provide also for table 3 the absolute numbers and percentages (for all educational categories) of those vaccinated but differentiate the percentages by writing for this column ‘vaccinated n (row %)’. The p-values should be reported in the last column.

  We thank the observations; we applied the suggested methods.

25. Table 4: Change the title to: “Results of multiple logistic regression of the intention to get the flu vaccination in the season 2023/24 or having already received it on several dichotomized predictive factors (yes/no refers to their presence or absence, respectively)”. Omit the useless footnote and the indication (1-0) and add four columns with the following entries: Yes n|Yes % vaccinated|No n|No % vaccinated. The yes/no refers to the variable in the line (e.g. Males). Report the CI in the form x-y and not in two columns. Include also age (below and above 60) and education (less or above high school level). The p-values should be reported in the last column.

 Done

26. In order to make this paper a meaningful contribution the following additional analyses should be done and reported. 1) A multivariate (or if your software cannot calculate it you can also do separate analyses for each of the dependent variables) logistic regression of the 5 main reasons not to get vaccinated (you may combine fear of side effects and the opinion the vaccine is dangerous, because this is more or less the same) on the same predictors as reported in Table 4. 2) A cluster analyses of the reasons to reject flu vaccination. There might be two or three patterns of opinions that may turn out to need different pathways and arguments to improve acceptance of vaccination.

We thank the observations, we applied the suggested methods.